# Sex-related differences in pre-dialysis trajectories and dialysis initiation: A French nationwide retrospective study

**Maxime Raffray** [1], **Louise Bourasseau**[1], **Cécile Vigneau**[2]*, **Cécile Couchoud**[3], **Clémence Béchade**[4], **François Glowacki**[5], **Sahar Bayat**[1], **on behalf of the REIN registry**¶

**1** Univ Rennes, EHESP, CNRS, Inserm, Arènes—UMR 6051, RSMS (Recherche sur les Services et Management en Santé), Rennes, France, **2** Univ Rennes, CHU Rennes, Inserm, EHESP, Irset (Institut de Recherche en Santé, Environnement et Travail), Rennes, France, **3** Renal Epidemiology and Information Network (REIN) Registry, Biomedicine Agency, Saint-Denis-La-Plaine, France, **4** Néphrologie, CHU Clemenceau, Caen, France, **5** Service de néphrologie, Hôpital Huriez, CHRU de Lille, Lille, France

¶ Membership of the REIN registry is provided in the Acknowledgments.
* cecile.vigneau@chu-rennes.fr

**Data Availability Statement:** The data used in the study cannot be shared publicly because it comprises of healthcare use at the individual level of French citizen (SNDS database). Data from the

## Abstract

### Background

In the last two decades, sex and gender differences have been documented in chronic kidney disease (CKD) management, including access to renal replacement therapy and its outcomes. The objectives of this study were to 1) compare the pre-dialysis healthcare utilization in men and women, and 2) examine the sex-specific factors associated with emergency dialysis start.

### Methods

Adult patients with CKD who started dialysis in France in 2015 were extracted from the Renal Epidemiology and Information Network registry. Patients were matched to the French National Health Data System database to extract healthcare utilization data for the 2 years before dialysis start. Frequencies and monthly rates of consultations and hospitalizations were compared between men and women. Logistic regression analyses were performed separately in the two groups.

### Results

Among the 8856 patients included, 3161 (35.7%) were women. Median age (71 years) and estimated glomerular filtration rate (8.1 and 7.7 ml/min for men and women) were similar between groups at dialysis start. Monthly consultations rates with a general practitioner and nephrology-related care were similar between women and men. Some sex-specific differences were found: higher frequencies of consultations with a psychiatrist in women and more frequent hospitalizations for circulatory system diseases in men. Emergency dialysis start rate was 30% in both groups. Emergency dialysis start was associated with acute nephropathy, compared with slowly progressive nephropathy, in women but not in men (OR = 1.48, p<0.01 vs 1.15, p = 0.18).

SNDS are available from the French Health Data Hub (https://smex-ctp.trendmicro.com:443/wis/clicktime/v1/query?url=https%3a%2f%2fwww.health%2ddata%2dhub.fr%2fdepot&umid=cbfb002f-8daa-401b-b5fe-59ff4db6872a&auth=cb5fd73bba5cd044cb5e5d90152ce147860ea990-d41f3f7e4e4e3a1074ee8ec2f11c75ea6e0d08ca) for researchers who meet the criteria for access to confidential data. The process will require authorization from the French Data Protection Authority (Commission nationale de l'informatique et des libertés). For this particular study, access was granted by the French Data Protection Authority (Commission nationale de l'informatique et des libertés, CNIL authorization N° 917021). Additionally, the REIN registry data was linked with the SNDS data with the approval of the relevant French committees, the Comité consultatif sur le traitement de l'information en matière de recherche (CCTIRS) and the French Data Protection Authority (Commission nationale de l'informatique et des libertés) (CNIL N° 903188). Information about the data of the REIN registry can be requested by mail to Dr. Cécile Couchoud who coordinates the REIN registry at the French Biomedicine Agency (cecile.couchoud@biomedecine.fr). We have joined the authorizations cited during the submission process.

**Funding:** This research was funded by a PhD grant from University of Rennes 1, France. The funders had no role in study design, data collection and analysis, decision to publish, or preparation of the manuscript.

**Competing interests:** The authors have declared that no competing interests exist.

## Conclusions

This study found similar quantitative pre-dialysis healthcare utilization in men and women. To better understand sex/gender differences in CKD care trajectories, future research should focus on patients with CKD who are unknown to nephrology services, on patients receiving conservative care and on the sex/gender-specific mechanisms underlying care decision-making.

## Introduction

Sex (biological) and gender (sociocultural construct)-specific differences have been documented for many diseases and various health-related areas (e.g. epidemiology, physiopathology, response to treatment and outcomes) [1]. Sex/gender-related differences and disparities also concern the interaction with healthcare, including access and utilization [2–6]. Chronic Kidney Disease (CKD) is not exempt from sex/gender-specific differences [7, 8]. In many countries, prevalence estimations show higher proportion of women with stage 3 to 5 CKD compared with men [7, 9–14]. However, more men start renal replacement therapy, including dialysis [15], possibly due to faster CKD progression [16–18], higher mortality rate in women, or disparity in its access [19].

In the last two decades, sex-related differences have been documented throughout CKD care including hemodialysis outcomes and access to transplantation [7, 20]. Women receiving hemodialysis have poorer health-related quality of life [21, 22], higher symptom burden and severity [23–25], higher hospitalization rates [26] and higher withdrawal risk [27]. Women also are less likely to being referred [28] and waitlisted for transplantation [29–31], and to receive a kidney transplant [32, 33].

Knowledge is limited on the healthcare utilization and trajectories leading to renal replacement therapy. In France, dialysis is provided by both public and private facilities. Dialysis treatment is 100% covered by social security and thus ownership of the facility should not be associated with its access. However, it is not known whether and how care trajectories differ between men and women before dialysis initiation.

Additionally, Emergency Dialysis Start (EDS) remains frequent in patients with CKD [34–36] and the sex/gender-specific risk factors of EDS have not been explored. Understanding these potential differences and disparities could contribute to targeted and personalized interventions and could improve the care of men and women with CKD.

Therefore, in this study, we wanted to describe and compare in men and women the pre-dialysis healthcare utilization in the last 2 years before dialysis initiation. We also examined the sex-specific factors associated with EDS.

## Materials and methods

### 1) Population studied and data collected

Patients and their individual data were collected from two French national databases. First, all adult patients who started dialysis in 2015 were identified in the Renal Epidemiology and Information Network (REIN) registry. This database records all patients with CKD who start renal replacement therapy in France and follows their care trajectories (e.g. waitlisting and transplant, dialysis withdrawal, death) [37, 38]. The REIN registry collects socio-demographic, clinical and laboratory data, initial kidney disease and comorbidities, renal replacement

therapy start modalities and context, including EDS. EDS is defined as a first dialysis initiated in the 24 hours following the identification of life threatening risks by a nephrologist. Three nephrologists (including the authors CV and CC) reviewed and classified the initial kidney diseases of the included patients in three groups according to their usual progression rate: slowly progressive, acute, and uncertain/variable nephropathy (S1 Table in S1 File).

As the REIN registry does not contain data on the pre-dialysis period, a deterministic record linkage was done with the French National Health Data System (SNDS), described elsewhere [39]. The SNDS is a nationwide database that covers 99% of the French population and contains data on a) all reimbursement of non-hospital-based outpatient healthcare (e.g. consultations, laboratory tests, and drug deliveries) and b) all hospital activity (i.e. inpatient and outpatient stays, diagnoses, procedures, and length of stay) [40]. However, it does not contain the results of laboratory tests (e.g. blood creatinine). The deterministic record linkage allowed extracting from the SNDS database healthcare data for the 2 years before dialysis start. The 2 years window was chosen and deemed appropriate for studying EDS based on the French CKD management guidelines recommending starting preparation for dialysis at least 1 year before its foreseeable start [41]. For the care trajectories analyses, healthcare data on consultations with all medical specialties, blood creatinine measurements, and hospitalizations were used [42]. REIN registry patients who could not be matched in the SNDS database and patients without information on dialysis start context (emergency or planned) were excluded.

## 2) Ethics statement

The REIN registry was approved by the relevant French committees, the Comité consultatif sur le traitement de l'information en matière de recherche (CCTIRS) and the Commission nationale de l'informatique et des libertés (CNIL N˚ 903188). Patients are informed about the registration in the REIN registry and their right to not participate (opt out) by the nephrology clinic.

Additionally, the permission to access the raw data used in our study (the SNDS data linked with the REIN registry data) was granted by the French Data Protection Authority (Commission Nationale Informatique et Liberté, CNIL authorization N˚ 917021). All data used in this study was anonymized before its use. Data was accessed between 10/01/2018 and 12/31/2021. All methods in this study were performed in accordance with relevant guidelines and regulations.

## 3) Analyses

Characteristics at dialysis start were described and compared between men and women. As the study population was not a sample, but the dialysis incident population in 2015, absolute differences and not p-values were used for comparison between groups (i.e. women and men).

Consultations were classified depending on the medical specialty: general practitioner (GP), nephrologist, and all other medical specialties. Hospitalizations were categorized in function of the main medical reason of admission (main diagnosis, International Classification of Diseases-10 codes; S2 Table in S1 File) and duration: a) hospitalizations related to dialysis preparation, b) <24-hour hospitalizations (all diagnoses), c) ≥24-hour hospitalizations related to nephrology, and d) ≥24-hour hospitalizations unrelated to nephrology. To describe the changes in healthcare utilization over time, incidence densities (or monthly consultation and hospitalization incidence rates) were calculated and plotted for the 24 months before dialysis start. Analyses were also stratified by age (<60 and ≥60 years), but no sex difference was observed.

Logistic regression analyses were performed separately in men and women to identify sex-specific factors associated with EDS. Regression models included the patients' characteristics at dialysis start (REIN registry) and pre-dialysis healthcare utilization (SNDS). Variables with a univariate p-value <0.20 were included in the multivariable analysis. A p-value <0.05 was considered statistically significant. Missing data were considered missing at random and imputed using the multiple imputation by chained equation procedure with ten iterations resulting in five imputed datasets [43].

## Results

### 1) Patients' characteristics

In total, 8856 patients were included in the analysis (Fig 1), among whom 3161 (35.7%) were women. Table 1 presents their characteristics at dialysis start. Polycystic kidney disease was more frequent among women, while vascular and hypertensive nephropathies were more frequent in men who also had more cardiovascular diseases (17.8% of men had ≥3 cardiovascular diseases vs 9% of women). Chronic respiratory disease also was more frequent in men (15.6% versus 9.7%). Obesity (body mass index ≥30 kg/m$^2$) was more frequent in women than men (25.5% versus 17.8% in men). The estimated glomerular filtration rate (eGFR) at dialysis start was similar between groups (median value: 8.3 ml/min in men and 7.7 ml/min in women) as well as EDS rate (30.9% in men and 29.1% in women).

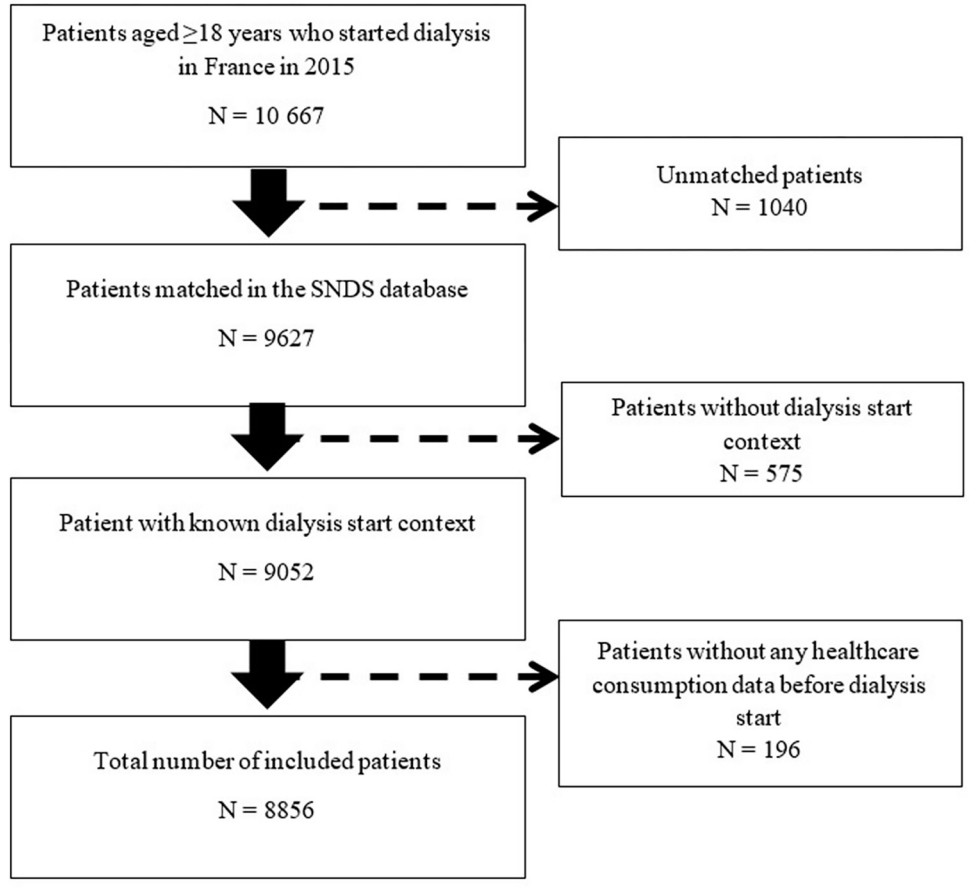

**Fig 1. Flowchart of patient inclusion.**

**Table 1. Characteristics at dialysis initiation (n = 8856 incident patients in France in 2015).**

| | Men | Women | Absolute difference (Women-Men) |
|---|---|---|---|
| | N = 5695 (63.4%) | N = 3161 (35.7%) | |
| **Age, median [Q1-Q3]** | 71.0 [60.9, 80.0] | 71.3 [59.9, 80.4] | 0.3 |
| 18–45 | 467 (8.2%) | 269 (8.5%) | 0.3 |
| 45–60 | 865 (15.2%) | 523 (16.5%) | 1.3 |
| 60–75 | 2094 (36.8%) | 1064 (33.7%) | -3.1 |
| ≥ 75 | 2269 (39.8%) | 1305 (41.3%) | 1.5 |
| **Initial kidney disease (nephropathy)** | | | |
| Polycystic | 232 (4.1%) | 220 (7.0%) | 2.9 |
| Diabetic | 1310 (23.0%) | 754 (23.9%) | 0.9 |
| Vascular and hypertensive | 1630 (28.6%) | 748 (23.7%) | -4.9 |
| Glomerulonephritis | 747 (13.1%) | 334 (10.6%) | -2.5 |
| Pyelonephritis | 232 (4.1%) | 135 (4.3%) | 0.2 |
| Others and unknown | 1544 (27.1%) | 970 (30.7%) | 3.6 |
| **Nephropathy progression type** | | | |
| Slowly progressive | 3900 (68.5%) | 2063 (65.3%) | -3.2 |
| Acute | 641 (11.3%) | 435 (13.8%) | 2.5 |
| Variable/uncertain | 1154 (20.3%) | 663 (21.0%) | 0.7 |
| **Body mass index (kg/m$^2$)** | | | |
| < 18.5 | 120 (2.1%) | 123 (3.9%) | 1.8 |
| 18.5–23 | 1170 (20.5%) | 622 (19.7%) | -0.8 |
| 23–25 | 811 (14.2%) | 341 (10.8%) | -3.4 |
| 25–30 | 1651 (29.0%) | 718 (22.7%) | -6.3 |
| ≥ 30 | 1012 (17.8%) | 807 (25.5%) | 7.7 |
| Missing | 931 (16.3%) | 550 (17.4%) | 1.1 |
| **Serum albumin (g/L)** | | | |
| ≥ 30 | 3985 (70.0%) | 2144 (67.8%) | -2.2 |
| < 30 | 987 (17.3%) | 570 (18.0%) | 0.7 |
| Missing | 723 (12.7%) | 447 (14.1%) | 1.4 |
| **Hemoglobin (g/L)** | | | |
| < 10 | 2974 (52.2%) | 1786 (56.5%) | 4.3 |
| 10–12 | 1698 (29.8%) | 899 (28.4%) | -1.4 |
| ≥ 12 | 818 (14.4%) | 350 (11.1%) | -3.3 |
| Missing | 205 (3.6%) | 126 (4.0%) | 0.4 |
| **Number of cardiovascular diseases*** | | | |
| 0 | 2265 (39.8%) | 1678 (53.1%) | 13.3 |
| 1 | 1401 (24.6%) | 757 (23.9%) | -0.7 |
| 2 | 1015 (17.8%) | 442 (14.0%) | -3.8 |
| ≥ 3 | 1014 (17.8%) | 284 (9.0%) | -8.8 |
| **Diabetes** | 2585 (45.4%) | 1401 (44.3%) | -1.1 |
| Missing | 26 (0.5%) | 14 (0.4%) | -0.1 |
| **Chronic respiratory disease** | 886 (15.6%) | 306 (9.7%) | -5.9 |
| Missing | 173 (3.0%) | 87 (2.8%) | -0.2 |
| **Mobility** | | | |
| Walk without help | 4438 (77.9%) | 2358 (74.6%) | -3.3 |
| Need assistance | 561 (9.9%) | 402 (12.7%) | 2.8 |
| Totally dependent | 225 (4.0%) | 154 (4.9%) | 0.9 |
| Missing | 471 (8.3%) | 247 (7.8%) | -0.5 |

(*Continued*)

**Table 1.** (Continued)

|  | Men | Women | Absolute difference (Women-Men) |
|---|---|---|---|
|  | **N = 5695 (63.4%)** | **N = 3161 (35.7%)** |  |
| **Emergency dialysis start** | 1762 (30.9%) | 919 (29.1%) | -1.8 |
| **Dialysis modality** |  |  |  |
| Hemodialysis | 5188 (91.1%) | 2856 (90.4%) | -0.7 |
| Peritoneal dialysis | 507 (8.9%) | 305 (9.6%) | 0.7 |
| **eGFR (ml/min per 1.73m², CKD-EPI) Median [Q1, Q3]** | 8.3 [6.1, 11.1] | 7.7 [5.7, 10.1] | -0.6 |
| Missing | 546 (9.6%) | 321 (10.2%) | -0.6 |

*Congestive heart failure, coronary disease, arrhythmia, aortic aneurysm, arteritis of lower limbs, stroke, or transient ischemic attack.

## 2) Comparison of the 2-year pre-dialysis care trajectories

During the 2 years before dialysis start, non-hospital care utilization was similar between men and women (Table 2). Specifically, the median number of consultations with a nephrologist was five and the median number of blood creatinine measurements was 24 in both groups. Overall, the percentage of patients with at least one nephrologist consultation increased during the study period from 43% of men and 40.8% of women in the fourth semester to 71.2% of men and 70.5% of women in the last semester before dialysis start. Women and men had a median number of 15 and 14 consultations with a GP, respectively, during the 2 years before dialysis start. Although changes over time were similar between groups, the monthly incidence rates of consultations with a GP (Fig 2) were consistently higher for women than for men (73.8 consultations per patient month versus 65.1).

The monthly incidence rates of consultations with other medical specialists were similar between men and women (Fig 2), but some differences were observed concerning the specialties (Table 3). Psychiatrists represented the sixth specialist consulted by women (6.2% of all consultations versus 2.7% for men). Conversely, urologic surgeons were the third specialist most frequently seen by men (8.9% of consultations) and the tenth by women (4% of consultations). Vascular surgeons and cardiologist also were more frequently consulted by men. The median number of different specialists seen at least once during the 2 years before dialysis start (excluding GP and nephrologist) was 3 in both groups. Detailed results on other specialists are presented in S3 Table in S1 File.

Overall, inpatient care utilization was similar between men and women. In both groups, nephrology-related hospitalizations increased in the last 6 months before dialysis start (reaching 28 hospitalizations per 100 patient-month in the last month). Conversely, hospitalizations unrelated to nephrology (all other causes) remained stable in the 2 years before dialysis start (median of 3.5 hospitalizations per 100 patient-month) (Fig 3). The proportion of patients who had ≥2 hospitalizations related to dialysis preparation was higher in women than men (15.6% versus 11.2%). However, the monthly incidence rate overlapped between groups (Fig 3). The median hospital stay was the same in men and women (6 and 3 days for hospitalizations related and unrelated to nephrology, respectively). Hospitalizations for <24 hours were slightly more frequent in women (Table 2), especially in the last 7 months before dialysis start (median of 9 and 7.5 hospitalizations per 100 patient-month) (Fig 3).

Among the main reasons of nephrology-unrelated hospitalizations, circulatory system diseases and cancer were more frequent among men (29.3% vs 24% and 13.7% vs 10.4% of those hospitalizations for men and women, respectively) (Fig 4). Conversely, injury/poisoning and

**Table 2. Healthcare utilization in the 2 years before dialysis start by sex (n = 8856 patients who started dialysis in France in 2015).**

|  | Men | Women |
|---|---|---|
|  | (N = 5695) | (N = 3161) |
| **General practitioner** |  |  |
| Median number of consultations [Q1, Q3] | 14.0 [8, 23] | 15.0 [9, 25] |
| No consultation during the last year before dialysis | 421 (7.4%) | 198 (6.3%) |
| ≥ 1 consultation during the 4th semester before dialysis start | 4783 (84.0%) | 2677 (84.7%) |
| ≥ 1 consultation during the 3rd semester before dialysis start | 4814 (84.5%) | 2713 (85.8%) |
| ≥ 1 consultation during the 2nd semester before dialysis start | 4884 (85.8%) | 2727 (86.3%) |
| ≥ 1 consultation during the 1st semester before dialysis start | 5020 (88.1%) | 2825 (89.4%) |
| **Nephrologist** |  |  |
| Median number of consultations [Q1, Q3] | 5.00 [1, 8] | 5.00 [1, 8] |
| No consultation during the last year before dialysis | 1387 (24.4%) | 796 (25.2%) |
| ≥ 1 consultation during the 4th semester before dialysis start | 2446 (43.0%) | 1290 (40.8%) |
| ≥ 1 consultation during the 3rd semester before dialysis start | 2780 (48.8%) | 1504 (47.6%) |
| ≥ 1 consultation during the 2nd semester before dialysis start | 3274 (57.5%) | 1783 (56.4%) |
| ≥ 1 consultation during the 1st semester before dialysis start | 4053 (71.2%) | 2228 (70.5%) |
| **Blood creatinine** |  |  |
| Median number of measurements [Q1, Q3] | 24.0 [10, 36] | 24.0 [10, 36] |
| **Hospitalizations** |  |  |
| Number of hospitalizations related to nephrology (≥24 h) |  |  |
| 0 | 1920 (33.7%) | 1067 (33.8%) |
| 1 | 1136 (19.9%) | 596 (18.9%) |
| ≥2 | 2639 (46.3%) | 1498 (47.4%) |
| *Median duration in days [Q1, Q3]* | *6 [3, 10]* | *6 [3, 11]* |
| Number of hospitalizations unrelated to nephrology (≥24 h) |  |  |
| 0 | 3703 (65.0%) | 2080 (65.8%) |
| 1 | 949 (16.7%) | 527 (16.7%) |
| ≥2 | 1043 (18.3%) | 554 (17.5%) |
| *Median duration in days [Q1, Q3]* | *3 [1, 7]* | *3 [1, 8]* |
| Number of hospitalizations related to dialysis preparation care |  |  |
| 0 | 3020 (53.0%) | 1654 (52.3%) |
| 1 | 2040 (35.8%) | 1015 (32.1%) |
| ≥2 | 635 (11.2%) | 492 (15.6%) |
| *Median duration in days [Q1, Q3]* | *2 [0, 2]* | *2 [0, 3]* |
| Number of hospitalizations <24 h |  |  |
| 0 | 1787 (31.4%) | 992 (31.4%) |
| 1 | 1558 (27.4%) | 770 (24.4%) |
| ≥2 | 2350 (41.3%) | 1399 (44.3%) |

endocrine/nutritional/metabolic diseases were more frequent reasons of hospitalization among women (7.2% vs 4.4% and 7.1% vs 3.9% of hospitalizations for women and men, respectively) (Fig 4).

## 3) Sex-specific factors of emergency dialysis start

The multivariable logistic regressions performed separately for men and women (Table 4) showed that compared with the 18–44 years group, EDS risk was lower in all the other age groups (45–59, 60–74, ≥75 years groups) in men (OR = 0.71, p = 0.01; OR = 0.65, p value;

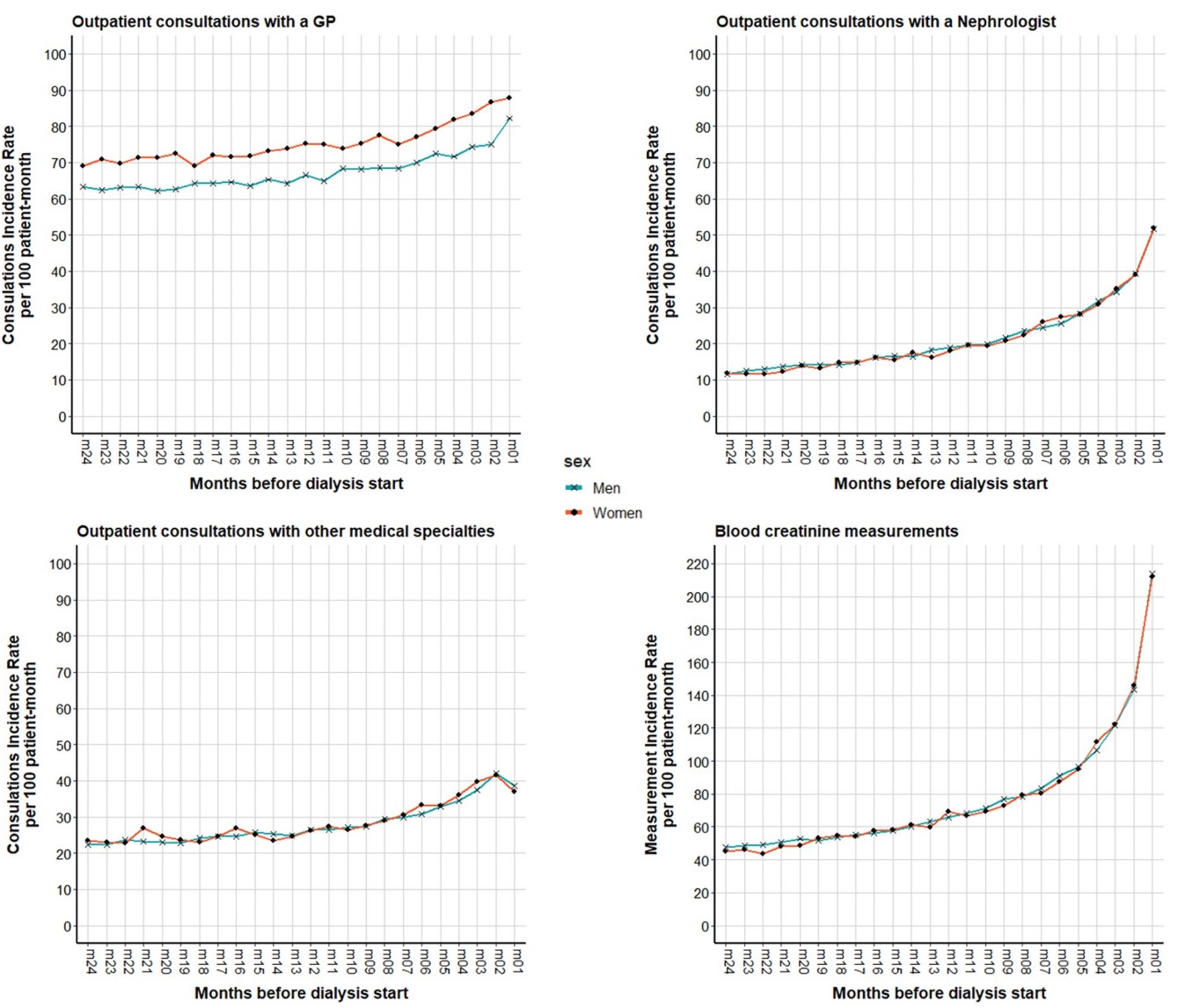

**Fig 2. Incidence rates per patient-month of non-hospital care by sex (N = 5695 men and 3161 women) in the two years before dialysis start.**

OR = 0.57, p<0.01, respectively), but only in the ≥75 years group in women (OR = 0.69, p = 0.04). Chronic respiratory disease was associated with a 30% increase in the risk of EDS among men (p<0.01), but not in women (p = 0.84). Needing assistance to walk and being totally dependent were risk factors of EDS in men, but not in women. Conversely, active malignancy was significantly associated with a 36% risk increase of EDS (p = 0.04) in women, but not in men (p = 0.08). Having an acute nephropathy, compared to a slowly progressive nephropathy, was associated with an increased risk of EDS in women (OR = 1.48, p<0.01), but not in men (OR = 1.15, p = 0.18).

Having seen a nephrologist at least three times in the year before dialysis start, compared with once or twice, was associated with a reduced risk of EDS in both men and women (OR = 0.58 and 0.56, p<0.01 respectively). One nephrology-related hospitalization, compared to none, was associated with a decreased risk of EDS only in men (OR = 0.81 p = 0.01). Two or more <24-hour hospitalizations also were associated with a decreased EDS risk in men, but not in women (OR = 0.75, p<0.01, OR = 0.81, p = 0.09).

**Table 3. Consultations with other medical specialties in the 2 years before dialysis start (top 10 most frequent).**

| Medical specialty | Men (N = 5695) | | Medical specialty | Women (N = 3161) | |
|---|---|---|---|---|---|
| | Number of consultations (total = 38162) | | | Number of consultations (total = 21505) | |
| | n | % | | n | % |
| Anesthesiology | 3933 | 10.3 | Anesthesiology | 2208 | 10.3 |
| Ophthalmology | 3731 | 9.8 | Ophthalmology | 2202 | 10.2 |
| Urologic surgery | 3379 | 8.9 | Endocrinology | 1621 | 7.5 |
| Endocrinology | 3081 | 8.1 | Vascular surgery | 1480 | 6.9 |
| Vascular surgery | 2965 | 7.8 | Cardiology | 1382 | 6.4 |
| Cardiology | 2855 | 7.5 | Psychiatry | 1341 | 6.2 |
| General surgery | 2028 | 5.3 | General surgery | 1223 | 5.7 |
| Internal Medicine | 1846 | 4.8 | Internal medicine | 1077 | 5.0 |
| Dermatology | 1664 | 4.4 | Orthopedic surgery and traumatology | 878 | 4.1 |
| Dental surgery | 1468 | 3.8 | Urologic surgery | 858 | 4.0 |

The cumulative top 10 most frequent specialties represented 70.6% and 66.4% of the total consultations with other medical specialties for men and women, respectively.

Reading example: 10.2% of the total consultations with other medical specialties for women were related to ophthalmology.

## Discussion

In this study, we examined and compared the care trajectories of men and women with CKD who started dialysis in France in 2015. We found that despite clinical characteristic differences at dialysis start, overall healthcare utilization in the previous 2 years was similar between sexes, including outpatient and inpatient care.

First, the temporal trends of pre-dialysis consultations with a GP were similar between sexes, although women had slightly more visits (median consultations: 15 vs 14). Women seeking GPs for reproductive related consultations could explain this slight difference [44]. It should be noted that compared with other studies, our population of patients with pre-dialysis CKD was older (median age: 71 years). This is important because a previous work in the United Kingdom found a gender gap in primary care consultation mainly in the 16 to 60-year-old population (i.e. higher consultation rate by women) that tended to disappear in >60-year-old patients [44].

Second, for nephrology-related care, no difference was observed in temporal trends and number of consultations with a nephrologist, creatinine measurements and hospitalizations for dialysis preparation. This is a positive result in the context of inequalities in access to care, especially when lower awareness of CKD among women has been reported [45]. Our study presents quantitative findings (i.e. rates over time and median number of consultations); however, it did not investigate what happens during these consultations and the resulting therapeutic strategies (i.e. the shared medical decision process). Practices of physicians might differ depending on the patient's gender [46–48]. For example, in a recent qualitative study, nephrologists recognized that they were "*more inclined to suggest non-dialytic supportive management*" to women deemed "*too frail*" [49]. Moreover, we found that about 1 in 4 patients (both sexes) did not see a nephrologist in the year before dialysis. This remains an important problem to address and for which the causes might differ between genders.

This overall absence of differences between sexes contrasts with previous studies that reported many differences at various steps of the CKD care trajectories, often unfavorable to women [7]. In the current academic context, studies reporting significant differences are more likely to be published [50]. However, this might lead to a partial view of the problem of sex/gender inequalities, as stressed by Hecking et al. in a recent review on gender and CKD care

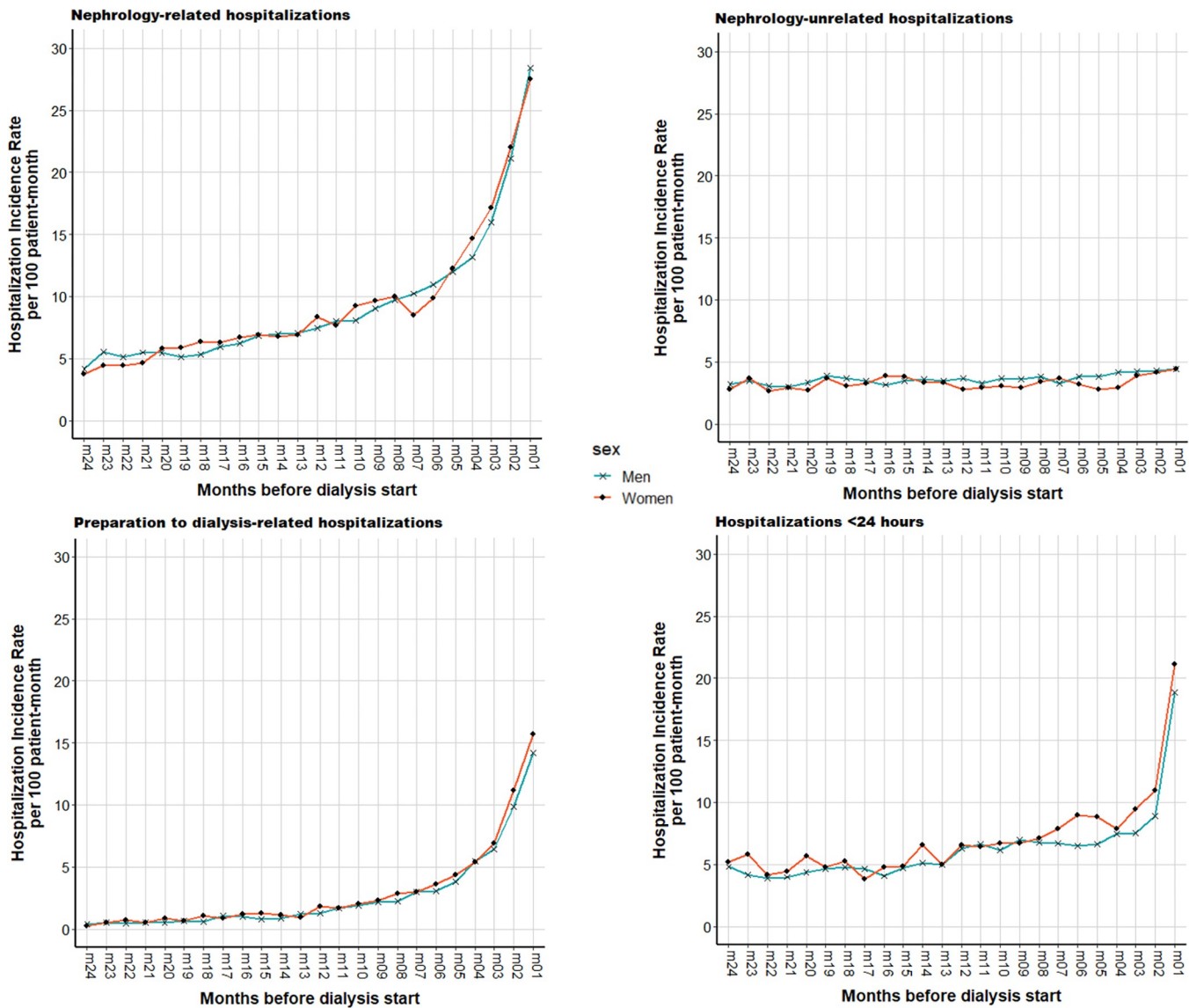

**Fig 3. Incidence rates per patient-month of hospitalizations by sex (N = 5695 men and 3161 women) in the two years before dialysis start.**

[51]: "*less attention is paid to findings in which women are better off or similar to men*". Therefore, our results are important and help to better understand the issue of sex/gender and CKD care, especially pre-dialysis care that has not been much investigated yet.

Nevertheless, our study also found some differences in the pre-dialysis care trajectories, particularly the higher number of consultations with a psychiatrist among women. This is consistent with literature data showing that in patients with CKD (pre-dialysis), psychiatric diseases are more frequent in women and depression is diagnosed less often in men [15, 52]. This difference in care utilization might be explained by how men and women react and cope with CKD. Indeed, men might find more difficult to reach out or accept to be referred to mental health specialists due to social pressure and perceived expectations. They may feel that they must display "*strength in silence*" [49, 53, 54]. Compared with men, women with CKD have lower risks of cardiovascular events and mortality [55]. Biological sex is an important modifier

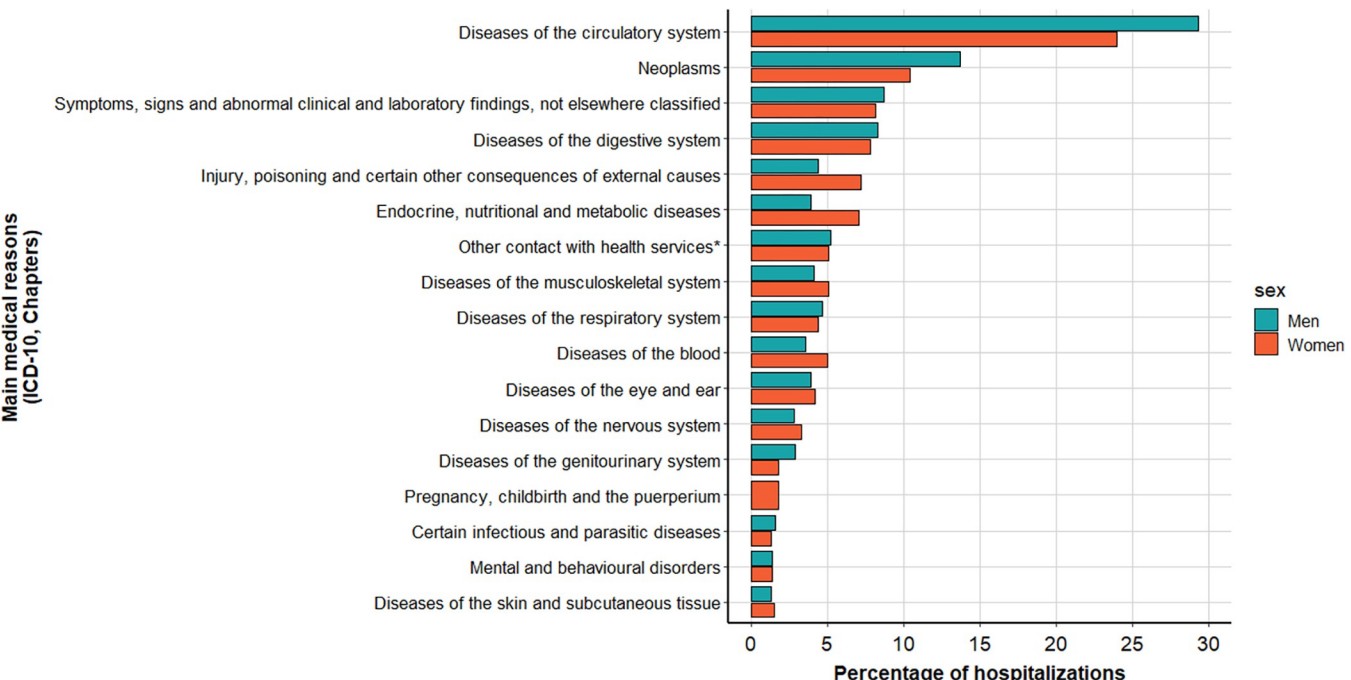

**Fig 4. Main medical reasons of nephrology-unrelated hospitalizations (≥24h) in the two years before dialysis start, for men (4058 hospitalizations for n = 1992) and women (2048 hospitalizations for n = 108)\*.** Adjustment and management of implanted device (e.g. cardiac devices), other surgical follow-up care, examination and observation for other reasons.

of the development of cardiovascular diseases. In particular, estrogen is considered as protective factor [56]. In our study, circulatory system diseases were the most frequent reason for consultation in both sexes, but were a more frequent reason of pre-dialysis hospitalization among men. This was consistent with the higher burden of cardiovascular comorbidities found in men at dialysis start, compared with women (i.e. higher number of cardiovascular diseases and frequency of hypertensive and vascular nephropathy). Nevertheless, the proportion of women and men who started dialysis in emergency (i.e. <24 hours after a nephrologist assessment due to life threatening reasons) was the same (30%). This could be explained by a greater capacity by women to tolerate and cope with CKD consequences, leading to delayed dialysis start compared with men for whom dialysis start might be anticipated due to their greater comorbidity burden. The analysis adjusted for patients' characteristics at dialysis start and pre-dialysis care confirmed that in both groups, lack of nephrological care was an important risk factor of EDS. Among the clinical factors associated with EDS, having an acute nephropathy (e.g. toxic nephropathy, myeloma, extracapillary glomerulonephritis), compared with a slowly progressive nephropathy, significantly increased the risk of EDS in women (+48%, p <0.01), but not in men. The causal relationship with EDS is unclear, but this suggests that in women, acute nephropathies might be more difficult to manage or worsen more rapidly. This should be taken into account by nephrologists and care teams.

## Strengths

This study has several strengths. The first is the exhaustiveness of the population included (i.e. all patients in France who started dialysis) and the quality of the baseline data extracted from the REIN registry [37], a long-established nationwide registry developed and supported by a network of nephrologists and clinical research assistants [38]. Second, the use of the French

**Table 4. Factors associated with EDS in men (N = 5695) and women (N = 3161), multivariable logistic regression.**

| | Men | Women |
|---|---|---|
| **Variable** | **Odds Ratio [95% Confidence Interval]** | **Odds Ratio [95% Confidence Interval]** |
| **Age (vs 18–44 years)** | | |
| 45–59 years | 0.71 [0.54, 0.93] | 0.79 [0.55, 1.13] |
| 60–74 years | 0.65 [0.50, 0.84] | 0.71 [0.50, 0.99] |
| ≥ 75 years | 0.57 [0.44, 0.74] | 0.69 [0.49, 0.98] |
| **Nephropathy (vs slowly progressive)** | | |
| Acute | 1.15 [0.94, 1.41] | 1.48 [1.14, 1.92] |
| Uncertain/variable progression | 1.18 [1.01, 1.38] | 1.29 [1.04, 1.59] |
| **Body mass index (kg/m$^2$) (vs 23–25 kg/m$^2$)** | | |
| < 18.5 | 1.07 [0.71, 1.63] | 1.11 [0.68, 1.79] |
| [18.5–23] | 1.29 [1.02, 1.61] | 0.99 [0.69, 1.41] |
| [25–30] | 0.97 [0.80, 1.18] | 0.96 [0.69, 1.35] |
| ≥ 30 | 1.03 [0.82, 1.29] | 0.90 [0.62, 1.30] |
| **Serum albumin < 30g/L (vs ≥30 g/L)** | 1.52 [1.26, 1.83] | 1.37 [1.10, 1.71] |
| **Hemoglobin (vs 10–12 g/dL)** | | |
| < 10 g/dL | 1.51 [1.31, 1.75] | 1.59 [1.30, 1.95] |
| ≥ 12 g/dL | 0.86 [0.69, 1.07] | 0.90 [0.64, 1.27] |
| **Cardiovascular diseases* (vs none)** | | |
| 1 | 1.18 [0.99, 1.40] | 1.31 [1.06, 1.63] |
| 2 | 1.53 [1.27, 1.85] | 1.42 [1.09, 1.86] |
| ≥ 3 | 1.81 [1.49, 2.20] | 1.99 [1.46, 2.72] |
| **Diabetes** | 1.20 [1.05, 1.38] | 1.26 [1.03, 1.53] |
| **Chronic respiratory disease** | 1.30 [1.10, 1.55] | 1.03 [0.78, 1.36] |
| **Active malignancy** | 1.20 [0.98, 1.46] | 1.36 [1.01, 1.82] |
| **Mobility (vs walk without help)** | | |
| Need assistance | 1.45 [1.19, 1.76] | 1.20 [0.92, 1.55] |
| Totally dependent | 1.65 [1.23, 2.23] | 1.49 [0.99, 2.23] |
| **Consultation with a GP (vs 1–7)**** | | |
| 0 | 1.09 [0.86, 1.38] | 0.81 [0.57, 1.16] |
| > 8 | 1.03 [0.90, 1.18] | 0.95 [0.79, 1.14] |
| **Consultations with a nephrologist (vs 1–2)**** | | |
| 0 | 1.47 [1.23, 1.75] | 1.71 [1.35, 2.17] |
| ≥ 3 | 0.58 [0.50, 0.68] | 0.56 [0.45, 0.70] |
| **Nephrology-related hospitalizations (vs 0)**** | | |
| 1 | 0.81 [0.68, 0.96] | 0.88 [0.69, 1.12] |
| ≥ 2 | 0.94 [0.81, 1.09] | 0.93 [0.76, 1.14] |
| **Nephrology-unrelated hospitalizations (all other causes) (vs 0)**** | | |
| 1 | 1.03 [0.86, 1.25] | 1.26 [0.99, 1.61] |
| ≥ 2 | 1.04 [0.84, 1.29] | 1.25 [0.94, 1.66] |
| **Preparation to dialysis-related hospitalizations (vs 0)**** | | |
| 1 | 0.40 [0.34, 0.47] | 0.44 [0.36, 0.55] |
| ≥ 2 | 0.36 [0.28, 0.46] | 0.49 [0.37, 0.67] |
| **<24 hours hospitalizations (vs 0)**** | | |
| 1 | 0.93 [0.79, 1.09] | 1.04 [0.83, 1.31] |

(*Continued*)

**Table 4.** (Continued)

| Variable | Men Odds Ratio [95% Confidence Interval] | Women Odds Ratio [95% Confidence Interval] |
|---|---|---|
| ≥ 2 | 0.75 [0.62, 0.90] | 0.81 [0.64, 1.04] |

*congestive heart failure, coronary disease, arrhythmia, aortic aneurysm, arteritis of lower limbs, stroke, or transient ischemic attack

National Health Data System allowed assessing the care trajectories, both non-hospital (e.g. consultations, laboratory tests, and drug deliveries) and hospital care, regardless of the providers' type (public or private sector) and the patients' insurance coverage type.

## Limits

The first and main limitation of this study is that the population studied only included the subset of patients with CKD who initiated dialysis. Sex and gender-based differences in the epidemiology and the care trajectories might exist among those who do not reach or choose renal replacement therapy. For example, evidence suggests that elderly women are more inclined to choose (or been offered) conservative care instead of renal replacement therapy, compared with men [7, 57]. In addition to prospective cohort studies, population-based studies that use healthcare claims databases might provide new insights on sex and gender differences in care trajectories. Indeed, identifying patients with CKD (any stage) through their healthcare utilization would allow including and studying patients unknown to the nephrology services and patients receiving conservative care [58]. Second, we could not assess some socio-economic factors, such as education, marital status and income, that might more strongly affect women than men [49, 59]. Third, as this study took place in France where universal healthcare coverage is offered, the results related to healthcare utilization might not be generalizable to other settings. Finally, although the data used date from 2015, we argue that is still of relevance today. Indeed, the CKD care management guidelines in France were last updated in 2021, with the notable introduction of nurses practitioners. The last previous update dates back from 2012. Although some improvements in CKD care management is reported by the REIN registry [60], some trends in the care of new dialysis patients have remained since 2015. We therefore argue that is unlikely that the main results of this study would be significantly different with more contemporary data.

## Conclusion

This study provides a quantitative description of the pre-dialysis care trajectories of women and men with CKD who started dialysis in France in 2015. Overall, healthcare utilization (GP and nephrological care) was comparable between sexes in the two years before dialysis initiation, despite differences in their clinical profiles at dialysis start. However, sex-specific differences were found for other healthcare areas, particularly higher frequencies of consultations with a psychiatrist among women and more frequent hospitalizations for circulatory system diseases among men. More quantitative and qualitative research is needed to comprehensively understand sex and gender differences in CKD care trajectories, specifically on patients with CKD unknown to nephrology services, on patients receiving conservative care, and on the decision-making processes.

## Supporting information

**S1 File.**
(PDF)

## Acknowledgments

The authors thank all REIN registry participants, especially nephrologists and data managers in charge of data collection and quality control: Nadia Honoré, Sabrina Boime, Emilie Gardeur-Algros, Observatoire régional de la santé du Grand Est Dr François Chantrel, Centre hospitalier, Mulhouse Xabina Larre, Dr Karen Leffondré, ISPED Bordeaux Dr Mathilde Reydit, AURAD; Eric Cellarier, Patricia Girault, CHU Clermont-Ferrand Aurélien Tiple, CHU Clermont-Ferrand; Aurélie Caillet, Dr Damiano Ceruasuolo, CHU Caen Dr Clémence Béchade, CHU Caen; Sophie Roche, Dr Anaïs Tendron-Franzin, CHU Dijon Dr Abdelkader Bemrah, Centre hospitalier Châlon/Saône; Muriel Siebert, CHU Rennes, Dr Sahar Bayat, EHESP, Rennes Pr Cécile Vigneau CHU Rennes; Marine Naudin, Dr Jean-Michel Halimi, CHU Tours Dr Bénédicte Sautenet, CHU Tours; Anne-Lise Varnier, Gwendoline Arnoult, Aurore Wolak, CHU Reims Dr Isabelle Kazès, CHU Reims; Ghizlane Izaaryene, Franck Mazoué, Adeline Cremades, Dr Stéphanie Gentile, CHU Marseille Dr Philippe Brunet, APHM Marseille; Caroline Savet, Dr Elisabeth Monnet, CHU Besançon Dr Cécile Courivaud, CHU Besançon; Cécilia Citadelle-Jannetta, Dr Jacqueline Deloumeaux, CHU Point-à-Pitre Dr Valérie Galantine, Clinique de Choisy, Gosier; Devi Rochemont, Mamadou Khali Sow, Pr Mathieu Nacher, Centre hospitalier Cayenne; Blandine Wurtz, Dr Hélène Marini, CHU Rouen Dr Stéphane Edet, CHU Rouen; Evelyne Ducamp, Zoubair Cherquaoui, Hayet Baouche, Houssem Eddine Tebbakh, Pr Jean-Philippe Jais, LBIM, Necker, APHP Dr Lucile Mercadal, Hôpital de la Pitié Salpétrière, Paris; Mohamed Belkacemi, Yohan Duny, Mélanie Martin, Dr Jean-Pierre Daurès, Université Montpellier Pr Olivier Moranne, CHU Carémeau, Nîmes; Florence Glaudet, Pr Alain Vergnenègre, CHU Limoges Dr Fatouma Touré, CHU Limoges; Marie-Rita Monzel, Véronique Vogel, Marie-Line Erpelding, Philippe Melchior, Amandine Ziegler, CIC 1433 Épidémiologie Clinique, CHRU Nancy Dr Emmanuelle Laurain, CHRU Nancy; Aurélie Bideau, Natacha Neller, Dr Sylvie Merle, Observatoire régional de la Santé, Fort de France Dr Alex Ranlin, ATIR; Violaine Schmitt, Catherine Marimoutou, CHU de la Réunion Pr Henri Vacher Coponat, CHU de la Réunion; Sophie Lapalu, Ludivine Brun, Dr Benoît Lepage, CHU Toulouse Dr Nathalie Longlune, CHU Rangueil Toulouse; Sébastien Gomis, Carole Foulon, Dr Marc Hazzan, CHU Lille Dr François Glowacki, CHU Lille; Noemie Baroux, RESIR—Réseau de l'insuffisance rénale en Nouvelle-Calédonie, Nouméa Dr Jean-Michel Tivollier, Nouméa; Ghizlane Izaaryene, Franck Mazoué, Adeline Crémades, Pr Stéphanie Gentile, CHU Marseille Pr Philippe Brunet, CHU Marseille; Assia Hami, Jean Xavier Lemauft, Jean-Michel Nguyen, CHU Nantes Pr Maryvonne Hourmant, CHU Nantes; Dr Etienne Bérard, CHU Nice; Amélie Joly, CHU Amiens Dr Ayman Sarraj, Polyclinique Saint Côme, Compiegne; Fabien Duthe, Bénédicte Ayrault, CHU Poitiers Dr Marc Bauwens, CHU Poitiers; Marie Hélène Pierron, Papeete Dr Pascale Testevuide, Papeete; Violaine Schmitt, Catherine Marimoutou, CHU de la Réunion Pr Henri Vacher Coponat, CHU de la Réunion, site Sud; Sylvie Boyer, Agnès Mérono, Marie-Noëlle Guillermin, Hospices Civils de Lyon Dr Roula Galland, Calydial, Irigny; Assia Hami, Jean Xavier Lemauft, Jean-Michel Nguyen, CHU Nantes Dr François Babinet, Echo Le Mans; The REIN registry national coordination can be reached at cecile.couchoud@-biomedecine.fr. The dialysis centers participating in the registry are fully listed in the REIN annual report (https://www.agence-biomedecine.fr/Le-programme-REIN). The authors thank Elisabetta Andermarcher for reviewing the English language.

## Author Contributions

**Conceptualization:** Maxime Raffray, Cécile Vigneau, Sahar Bayat.

**Data curation:** Cécile Couchoud.

**Formal analysis:** Maxime Raffray, Louise Bourasseau.

**Funding acquisition:** Sahar Bayat.

**Investigation:** Maxime Raffray, Louise Bourasseau.

**Methodology:** Maxime Raffray, Cécile Vigneau, Cécile Couchoud, Sahar Bayat.

**Project administration:** Sahar Bayat.

**Supervision:** Cécile Vigneau, Sahar Bayat.

**Validation:** Cécile Vigneau, Cécile Couchoud, Clémence Béchade, François Glowacki, Sahar Bayat.

**Visualization:** Maxime Raffray.

**Writing – original draft:** Maxime Raffray, Louise Bourasseau.

**Writing – review & editing:** Cécile Vigneau, Cécile Couchoud, Clémence Béchade, François Glowacki, Sahar Bayat.

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
