## [Decision Letter · Decision Letter 0]

2 Jan 2024

PONE-D-23-33969Sex-related differences in pre-dialysis trajectories and dialysis initiation: a French nationwide retrospective studyPLOS ONE

Dear Dr. Raffray,

Thank you for submitting your manuscript to PLOS ONE. After careful consideration, we feel that it has merit but does not fully meet PLOS ONE’s publication criteria as it currently stands. Therefore, we invite you to submit a revised version of the manuscript that addresses the points raised during the review process.

We look forward to receiving your revised manuscript.

Kind regards,

Chikezie Hart Onwukwe

Academic Editor

PLOS ONE

Journal Requirements:

"Rennes 1 University PhD grant, France"

3. In this instance it seems there may be acceptable restrictions in place that prevent the public sharing of your minimal data. However, in line with our goal of ensuring long-term data availability to all interested researchers, PLOS’ Data Policy states that authors cannot be the sole named individuals responsible for ensuring data access (http://journals.plos.org/plosone/s/data-availability#loc-acceptable-data-sharing-methods).

4. One of the noted authors is a group or consortium [REIN registry]. In addition to naming the author group, please list the individual authors and affiliations within this group in the acknowledgments section of your manuscript. Please also indicate clearly a lead author for this group along with a contact email address.

Reviewers' comments:

Reviewer's Responses to Questions

**Comments to the Author**

1. Is the manuscript technically sound, and do the data support the conclusions?

Reviewer #1: Yes

Reviewer #2: Yes

Reviewer #3: Yes

Reviewer #4: Yes

2. Has the statistical analysis been performed appropriately and rigorously? 

Reviewer #1: Yes

Reviewer #2: Yes

Reviewer #3: Yes

Reviewer #4: Yes

3. Have the authors made all data underlying the findings in their manuscript fully available?

Reviewer #1: Yes

Reviewer #2: Yes

Reviewer #3: Yes

Reviewer #4: Yes

4. Is the manuscript presented in an intelligible fashion and written in standard English?

Reviewer #1: Yes

Reviewer #2: Yes

Reviewer #3: Yes

Reviewer #4: Yes

5. Review Comments to the Author

Reviewer #1: Thanks for all authors for the efforts. I have the following points:

1. It would be appropriate to shortly describe the dialysis services and simple stats in France in the Introduction. Any private practice and how much it affects access and patients management.

2. Authors described Sex and Gender well. Is there any data about transgender and LGBT patients. Did they included such cases and how much this affect the result? Any data about immigrants and minorities?

3. The authors need to explain why women were seeking GP more? They put their explanation later on line 228 while they were explaining the diference in seeking Psychiatric care. This needs more emphasis.

4. It may be appropriate to consider the effect of estrogen in reducing the risk of CV disease in women compared to men.

Reviewer #2: The study by Raffray et al. aimed to determine differences in predialysis healthcare utilization and trajectories as well as emergency dialysis start between women and men. Based on the combination of two large registries they were able to analyze more than 8'000 patients in France. For most parts, they found no relevant sex/gender differences for health care utilization in patients starting dialysis therapy.

Strengths:

- The topic is of relevance, as of today it remains unclear, why consistently more men than women are on dialysis. Unfortunately, this study does not answer the question either.

- The study design and the analyses of the data seem to be sound, and the conclusions are adequately supported by the results.

Limitations:

- The most important limitation of the study - as stated by the authors themselves - is that it only includes patients, which eventually ended up on dialysis. Although it was not the aim of the analysis to determine the fate of every patient with CKD in the long run, this, nevertheless, is the ultimate question that needs to be resolved.

- Similarly, the study design does not allow to determine the number of patients who die before reaching renal replacement therapy in an "intention-to-treat manner". Thus, theoretically, an overproportionally high number of women who died prematurelly might not be accounted for.

Suggestions:

- It would be interesting to differentiate the patients with EDS according to their vascular access, i.e. av fistula versus catheter, and the respective differences in sex/gender.

Minor comments

- Line 178 (The multivariate logistic regressions performed separately for men and women…): multivariate should be replaced by multivariable, as it refers to independent rather than dependent factors.

Reviewer #3: This is a second manuscript by the same group of authors to cover an important issue regarding the possible diversity of sex and gender in health care services. The first covered the post-dialysis period, and this manuscript covers the pre-dialysis period. I have no comments.

Reviewer #4: Thank you very much for submitting your manuscript.

The manuscript carries a significant message.

Please clarify whether all French individuals accessthe National healthcare system and are captured in the SNDS data system.

What about immigrants and their healthcare support system.

Authors need to clarify why only two years of pre dialysis data was included in the analysis and whether this time frame is enough to discriminate between acute presentation and slowly progressive CKD.

What is the relevance of the data of 2015 at the end of 2023?

It would have been better if the authors had looked into the pre-dialysis cohort and look into the gender difference in patients who needed dialysis but did not start dialysis.

6. PLOS authors have the option to publish the peer review history of their article (what does this mean?). If published, this will include your full peer review and any attached files.

Reviewer #1: No

Reviewer #2: No

Reviewer #3: **Yes: **AHMED AKL

Reviewer #4: **Yes: **Urmila Anandh

---

## [Author Response · Author response to Decision Letter 0]

1 Feb 2024

Journal Requirements:

We have made changes in the manuscript to meet PLOS ONE’s style requirements, including title change and figures citations. We also uploaded each figure separately. We also made changes to the title page and affiliations format in accordance with the journal requirements. 

"Rennes 1 University PhD grant, France"

The statement “The funders had no role in study design, data collection and analysis, decision to publish, or preparation of the manuscript." was added in the funding statement of the manuscript and in the cover letter. 

3. In this instance it seems there may be acceptable restrictions in place that prevent the public sharing of your minimal data. However, in line with our goal of ensuring long-term data availability to all interested researchers, PLOS’ Data Policy states that authors cannot be the sole named individuals responsible for ensuring data access (http://journals.plos.org/plosone/s/data-availability#loc-acceptable-data-sharing-methods).

We added to the data availability statement in the method section the link to a form that can be used to access data from the REIN registry. The request of data access is reviewed and approved by the scientific committee of the REIN registry which constitutes a durable point of contact. 

4. One of the noted authors is a group or consortium [REIN registry]. In addition to naming the author group, please list the individual authors and affiliations within this group in the acknowledgments section of your manuscript. Please also indicate clearly a lead author for this group along with a contact email address.

We have included the list of members of the REIN registry consortium in the acknowledgements section of the manuscript. In the same section we added the contact email address for the national coordinator of the REIN registry consortium. 

Captions for the supporting information files were included at the very end of the manuscript. We renamed the supporting information as recommended in the guideline. The in-text citation of supporting information was updated. 

We reviewed the reference list and have no changes to report. 

Reviewers' comments:

Reviewer #1: Thanks for all authors for the efforts. I have the following points:

1. It would be appropriate to shortly describe the dialysis services and simple stats in France in the Introduction. Any private practice and how much it affects access and patients management.

We thank the reviewer for this comment. We added in the introduction that in France, dialysis is provided by both public and private facilities. Dialysis treatment is 100% covered by social security and thus ownership of the facility should not be associated with its access.

2. Authors described Sex and Gender well. Is there any data about transgender and LGBT patients. Did they included such cases and how much this affect the result? Any data about immigrants and minorities?

The REIN registry does not include yet information on trans identity. Similarly, sexual orientation is not collected. It is therefore difficult to estimate an impact on the results. The REIN registry does not contain information on race/ethnicity either. 

3. The authors need to explain why women were seeking GP more? They put their explanation later on line 228 while they were explaining the diference in seeking Psychiatric care. This needs more emphasis.

We found a slight difference in the median number of consultation with GP (15 among women vs 14). We added in the discussion that this slight difference might be due to the fact that women of reproductive age seeks GP for reproduction related motives, which is what the study of Wang et al. highlights. 

4. It may be appropriate to consider the effect of estrogen in reducing the risk of CV disease in women compared to men.

We added in the discussion that biological sex and hormones are important modifiers in the development of CVD, in particular estrogen with a protective role. See Arnold AP, Cassis LA, Eghbali M, Reue K, Sandberg K. Sex Hormones and Sex Chromosomes Cause Sex Differences in the Development of Cardiovascular Diseases. Arterioscler Thromb Vasc Biol. 2017 May;37(5):746–56

Reviewer #2: The study by Raffray et al. aimed to determine differences in predialysis healthcare utilization and trajectories as well as emergency dialysis start between women and men. Based on the combination of two large registries they were able to analyze more than 8'000 patients in France. For most parts, they found no relevant sex/gender differences for health care utilization in patients starting dialysis therapy.

Strengths:

- The topic is of relevance, as of today it remains unclear, why consistently more men than women are on dialysis. Unfortunately, this study does not answer the question either.

- The study design and the analyses of the data seem to be sound, and the conclusions are adequately supported by the results.

Limitations:

- The most important limitation of the study - as stated by the authors themselves - is that it only includes patients, which eventually ended up on dialysis. Although it was not the aim of the analysis to determine the fate of every patient with CKD in the long run, this, nevertheless, is the ultimate question that needs to be resolved.

- Similarly, the study design does not allow to determine the number of patients who die before reaching renal replacement therapy in an "intention-to-treat manner". Thus, theoretically, an overproportionally high number of women who died prematurelly might not be accounted for.

Suggestions:

- It would be interesting to differentiate the patients with EDS according to their vascular access, i.e. av fistula versus catheter, and the respective differences in sex/gender.

We thank the reviewer for their comments and suggestions. This was done in a previous paper by our team with the same data and population. We looked at the care trajectories following dialysis start. 

Patients were differentiated according to the vascular access and sex/gender. No major difference was found. For information, planned dialysis with fistula (38% among women vs 40%), planned dialysis with catheter (27% vs 24%), emergency dialysis start with fistula (4% vs 5%), emergency dialysis start with catheter (25% vs 26%). 

Piveteau J, Raffray M, Couchoud C, Chatelet V, Vigneau C, Bayat S. Care trajectory differences in women and men with end-stage renal disease after dialysis initiation. PLOS ONE. 2023 Sep 14;18(9):e0289134.

Minor comments

- Line 178 (The multivariate logistic regressions performed separately for men and women…): multivariate should be replaced by multivariable, as it refers to independent rather than dependent factors.

We thank the reviewer for this correction. We replace the term multivariate to multivariable in all the instances of the manuscript. 

Reviewer #3: This is a second manuscript by the same group of authors to cover an important issue regarding the possible diversity of sex and gender in health care services. The first covered the post-dialysis period, and this manuscript covers the pre-dialysis period. I have no comments.

Reviewer #4: Thank you very much for submitting your manuscript.

The manuscript carries a significant message.

Please clarify whether all French individuals accessthe National healthcare system and are captured in the SNDS data system.

What about immigrants and their healthcare support system.

We thank the reviewer for their comments and suggestions. All French individuals are captured in the SNDS database. The French Healthcare system also supports immigrants who do not have the French citizenship, and thus a social security number, with the State Medical Assistance program (abbreviated as AME). Thus, they can be capture in the SNDS database. However we did not differentiate between the two in this study. 

Authors need to clarify why only two years of pre dialysis data was included in the analysis and whether this time frame is enough to discriminate between acute presentation and slowly progressive CKD.

We thank the reviewer for this suggestion. We included in the analysis 2 years of pre-dialysis healthcare data based on the French national CKD management guidelines. Those guidelines recommend starting preparation of dialysis 1 year before the foreseeable start of dialysis. Therefore, to study emergency dialysis start, we deemed appropriate to look at two years of data with the hypothesis that patients should have been prepared for dialysis 1 year before dialysis start. We added that justification in the methods section of the revised manuscript. 

The 2-years of pre-dialysis data from the SNDS cannot be used to characterize the clinical progression of CKD because of its lack of lab value data. 

However, we have the information related to the nephropathy as a clinical data from the REIN registry, which records every kidney replacement therapy start. From clinical expertise from co-authors we derived the “Nephropathy progression type” via the nephropathy code. This classification is included in S1 Table. We described the differences regarding the types of nephropathies between men and women in Table 1. 

What is the relevance of the data of 2015 at the end of 2023?

We thank the reviewer for this question. It is difficult to evaluate how much CKD care has improved between 2015 and today. 

However, we can argue that 2015 data is still relevant as of now and the insights it provide in this study is important. 

First, we can refer to the trends in the care of new ESRD patients provided by the REIN registry. The proportion of emergency dialysis start has not decreased significantly. The REIN registry reports proportions ranging from 31% in 2015 to 25% in 2019 and 27% in 2021 (most recent data available). The proportion of dialysis initiation with a catheter has not changed since 2015 (54% vs 56% in 2019). The median age of new dialysis patients remained 71 years old and the sex ratio men/women was 1.8 in 2015 and 1.9 in 2021. 

Moreover, the French national CKD management guidelines have been updated only in 2021, with the last update dating from 2012. We therefore argue that is unlikely that the main results of this study would be significantly different with more contemporary data. 

We added this argument in the limitations section of the manuscript. 

It would have been better if the authors had looked into the pre-dialysis cohort and look into the gender difference in patients who needed dialysis but did not start dialysis.

The population of patients with CKD who reach the clinical point of starting dialysis but do not actually start dialysis (aka Kidney failure with or without comprehensive conservative care) or dies before is of great interest. The design of this study does not allow investigating those patients. This is a limitation that we state in the discussion. 

Such investigation requires a cohort of patients identified as having CKD with information on the grade and progression of the disease, which the SNDS currently does not have. 

Still, it remains important to document the differences, or lack-of, in care trajectories among that population of patients treated by dialysis, which is the objective of this particular study.

---

## [Decision Letter · Decision Letter 1]

14 Feb 2024

Sex-related differences in pre-dialysis trajectories and dialysis initiation: a French nationwide retrospective study

PONE-D-23-33969R1

Dear Maxime Raffray,

We’re pleased to inform you that your manuscript has been judged scientifically suitable for publication and will be formally accepted for publication once it meets all outstanding technical requirements.

Kind regards,

Chikezie Hart Onwukwe

Academic Editor

PLOS ONE

Additional Editor Comments (optional):

Reviewers' comments:

Reviewer's Responses to Questions

**Comments to the Author**

1. If the authors have adequately addressed your comments raised in a previous round of review and you feel that this manuscript is now acceptable for publication, you may indicate that here to bypass the “Comments to the Author” section, enter your conflict of interest statement in the “Confidential to Editor” section, and submit your "Accept" recommendation.

Reviewer #1: All comments have been addressed

Reviewer #2: All comments have been addressed

Reviewer #4: All comments have been addressed

2. Is the manuscript technically sound, and do the data support the conclusions?

Reviewer #1: Yes

Reviewer #2: Yes

Reviewer #4: Yes

3. Has the statistical analysis been performed appropriately and rigorously? 

Reviewer #1: Yes

Reviewer #2: Yes

Reviewer #4: Yes

4. Have the authors made all data underlying the findings in their manuscript fully available?

Reviewer #1: Yes

Reviewer #2: Yes

Reviewer #4: Yes

5. Is the manuscript presented in an intelligible fashion and written in standard English?

Reviewer #1: Yes

Reviewer #2: Yes

Reviewer #4: Yes

6. Review Comments to the Author

Reviewer #1: (No Response)

Reviewer #2: (No Response)

Reviewer #4: Thank you for addressing all the comments appropriately.

Some of the responses need to be added in the discussion for added clarity.

7. PLOS authors have the option to publish the peer review history of their article (what does this mean?). If published, this will include your full peer review and any attached files.

Reviewer #1: No

Reviewer #2: No

Reviewer #4: **Yes: **Urmila Anandh

---

## [Editor Report · Acceptance letter]

18 Mar 2024

PONE-D-23-33969R1 

PLOS ONE

Dear Dr. Raffray, 

I'm pleased to inform you that your manuscript has been deemed suitable for publication in PLOS ONE. Congratulations! Your manuscript is now being handed over to our production team.

Kind regards, 

on behalf of

Dr. Chikezie Hart Onwukwe 

Academic Editor

PLOS ONE